# Entrepreneurial Leadership and Employees' Proactive Behaviour: Fortifying Self Determination Theory

**Muhammad Bilal [1,\*], Shafaq Chaudhry [1], Hina Amber [2], Muhammad Shahid [1], Shoaib Aslam [3,\*] and Khuram Shahzad [4]**

[1]  Lahore Business School, The University of Lahore, Lahore 54000, Pakistan; Shafaq.arif@lbs.uol.edu.pk (S.C.); shahid.numl204@gmail.com (M.S.)
[2]  Beaconhouse School System, Attock 43600, Pakistan; hinaamber.76798@bh.edu.pk
[3]  Department of Commerce, The Islamia University of Bahawalpur, Bahawalpur 63100, Pakistan
[4]  Northern Business School, Northern University Nowshera, Nowshera 24110, Pakistan; khuram.shahzad@northern.edu.pk
\*  Correspondence: shorty.107@hotmail.com (M.B.); shoaib.aslam@iub.edu.pk (S.A.)

**Abstract:** The prevailing pandemic (COVID-19) has increased socioeconomic problems and caused psychological distress due to work uncertainty, specifically in emerging economies. Small and medium enterprises (SMEs) in emerging economies have been severely affected. Particularly, work uncertainty is becoming a hindrance towards proactive work behaviour (PWB) that can be improved by an effective entrepreneurial leadership role and proactive personality attribute. Based on fortifying self-determination theory, this research answered the question to what extent proactive personality moderates the relationship between work uncertainty and PWB and strengthens the relationship between entrepreneurial leadership and PWB. To empirically examine the study's underlying theoretical framework, respondents were selected from SMEs working in Pakistan from the high-tech industry. Multisource data were accumulated from 420 workers and their leaders utilizing a two-wave, time-lagged research design. Conclusions revealed that entrepreneurial leadership first reduced individuals' work uncertainty, which in turn, led to enhanced proactive work behaviour of employees. Furthermore, the results revealed that work uncertainty mediates the relationship between entrepreneurial leadership and proactive work behaviour. Moreover, proactive personality moderates the link concerning work uncertainty and proactive work behaviour, such that this association is significant only when proactive personality is low. Additionally, the moderated mediation analysis indicated that less proactive people, compared with their extraordinarily proactive colleagues, trusted entrepreneurial leadership to be more proactive in the workplace. These findings have important implications to induce PWB among employees.

**Keywords:** entrepreneurial leadership; work uncertainty; proactive work behaviour; proactive personality; innovation; self-determination theory





## 1. Introduction

The challenging business situation has created intense competition among high-tech small and medium enterprises (SMEs) to nurture and survive. Globally, SMEs are pretty significant in supporting countries' GDP and the economy. In Pakistan, until the 1960s, only 22 families were running businesses. To break their monopoly, the idea of launching SMEs was conceived. SMEs are the backbone of developing countries such as Pakistan, contributing more than 40% to the total GDP. The novel COVID-19, which appeared in Wuhan, China, was broadcasted as an epidemic through the World Health Organization (WHO) and has become the main challenge of the decade [1,2]. Initially, very few states succeeded to control the virus, such as Pakistan, but the second wave of COVID-19 affected India most prominently.

A global decline, economic slumps, and recession in businesses are just some of the repercussions of the international pandemic [3]. Moreover, in present uncertain, dynamic, and competitive environments, firms striving to prosper may reconsider preferences to support their structure with scientific modifications [4,5]. In such a state, proactivity is recognized as a significant compelling foundation for the competitiveness, attainment, and survival of high-tech organizations [6,7]. However, in the current scenario of work uncertainty, it has become demanding for corporate leaders to inspire the followers to disregard the outdated method of task performance and apply their vigour to become proactive in their behaviours [8].

Due to COVID-19, the dynamic structure of current organizations has forced both practitioners and scholars to focus on such behaviour that is change-oriented, future-focused, and self-starting [9]. The proactivity of employees can achieve these future-oriented, self-initiated behaviours at the workplace. Individuals can be proactive across a variety of domains, such as voice [10], averting problems [11], actively seeking feedback [12], a personal initiative [13], modifying jobs [14], and taking charge [15]. Numerous types of proactive behaviours are self-initiated instead of sticking to the prescribed way of doing things. Under self-initiated behaviour, employees focused on changing them rather than upholding the status quo or accommodating change. Thus, proactive employees are future intensive rather than reactive [16].

Because of the epidemic, developing work behaviour and an environment that supports and promotes proactive behaviour frequently requires crucial organizational work practices such as human resource management practices [17], leadership, and distribution of authority and power in organizations [18]. Mostly, leadership plays a substantial role in influencing and persuading a follower's behaviour. Thus, for encouraging employees to be involved in opportunity exploration and utilization by inculcating proactive behaviour [19], a specific leadership style called entrepreneurial leadership (EL) may be needed. Among numerous contextual factors, leadership significantly influences an employee's proactivity [10]. Moreover, leaders have the resources and power to change situations and policies [20]. However, irrespective of the leadership's role in subordinates' behaviour, empirical investigations on the topic are still insufficient [21]. Therefore, the present research focuses on entrepreneurial leadership to investigate its role in encouraging followers' proactive behaviour.

The entrepreneur continuously hunts for change, reacts to it, and utilizes it as a prospect. Thus, entrepreneurial leaders promote and encourage discretionary behaviour (proactive, innovative, and organizational citizenship behaviour) by changing, responding to, motivating, and exploiting opportunities at work to improve performance [21]. Proactive work behaviour (PWB) leads to innovation [22]. There is growing interest in exploring antecedents and the mechanism through which proactive behaviour can be induced in the organization settings [23] to increase innovative performance. Studies also argue that it is obligatory to scrutinize entrepreneurial leadership outcomes that may enhance creative organizational performance [21].

Undoubtedly, proactive behaviour is recognized as an influential component in the contemporary business environment's trajectory to sustain competitive advantage. Theoretical frameworks, i.e., self-determination theory [24], underpinning the theory of this research, enable and equip leaders to generate a guiding philosophy for improving organizational performance [25] through proactive behaviours of employees. SDT posits that individuals become intrinsically motivated through the environment created by leaders, and entrepreneurial leaders are considered legitimate role models for shaping such a climate [26]. Entrepreneurial leaders create such an environment in which autonomous motivation implies thoughts, values, attitudes, and morally right behaviour and can influence the employees' perceptions about policies, procedures, and practices regarding a supportive climate and will put in extra efforts to decrease work uncertainty. The boundary conditioning impact of proactive personality weakens the association between entrepreneurial

leadership and PWB as entrepreneurial leaders focus on creating a conducive climate for the employees by promoting an autonomy-focused supportive environment [27,28].

Furthermore, the prevailing virus has brutally affected national and international markets. Nevertheless, enterprises around the world are suffering substantial waves of the COVID-19 attack on their businesses. Numerous enterprises are encountering countless issues having significant losses. We believe that small and medium-sized enterprises became the substantial target in the COVID-19 epidemic. SMEs generally do not retain adequate assets equated with huge enterprises, mainly managerial and financial, and are not ready for such disorders expected to go longer than anticipated [1,2]. Pakistani SMEs have faced worse repercussions of COVID-19 in the shape of the scarcity of things, reduced products demand and amenities, transport blockage, reduced sales and profits, restricted operations, lockdown, and worker downsizing. In addition, it has exaggerated the lifestyle of people and inversely affected people's health and increased work uncertainty. Furthermore, there are still no estimates of how many states will suffer the fourth surge of COVID-19, which is expected to be more fatal than the previous one. Thus, this study will help in finding solutions for how to reduce work uncertainty and stress by improving psychological wellbeing through job employment.

Considering the improbability of COVID-19, the present research examined the latent influence of entrepreneurial leadership on the proactive work behaviour of the employees in SMEs. The desired population for this study was the personnel of the SMEs of Pakistan. As research in this segment is ignored, this research attempts to fill the breach and explore novel prospects for the leaders [20]. This study examines how perceived work uncertainty meditates the relationship between entrepreneurial leadership and proactive work behaviour. Lastly, this research examines how proactive personality moderates the relationship between entrepreneurial leadership and proactive work behaviour and even between work uncertainty and proactive work behaviour. By investigating these relationships, we can comprehend the contextual factor-like entrepreneurial leadership that promotes the proactive work behaviours of employees. This research also identifies more about the personality factor (proactive personality) influencing individuals during the pandemic. The core aim of the study was to add literature on the SMEs' segment of Pakistan, as this is a most ignored and under-researched segment [29]. Moreover, this is one of the segments that is experiencing severe consequences from this pandemic. The current paper investigates the relationships of entrepreneurial leadership, work uncertainty, proactive personality, and proactive work behaviour in employees of SMEs.

## 2. Literature Review and Hypothesis Development

Self-determination theory offers a predominantly fertile theoretical perceptive for enlightening how proactivity is enthused. The self-determination theory can add to how proactivity is created. In self-determination theory, a chain of autonomous to measured enthusiasm is anticipated, with various underlying courses of practices and consequences [30,31]. At one extreme of this chain is intrinsic motivation, which implicates the involvement in self-interested fruitful behaviour. At the other extreme of the chain is extrinsic motivation, which comprises the beginning and continuation of behaviour by considering the significances of external rewards (such as salary and incentives). Between the continuums of these two ends, identified, integrated, and introjected motivation, which observed as autonomous than peripheral motivation, are not intrinsic. Proactive behaviour is change-oriented self-initiated behaviour that needs autonomy and competency [16]. The emotional state of flow generated from involvement in stimulating activities [32], as the yearning for the stream can subsequently provoke proactive endeavours. At last, a few kinds of proactivity (e.g., singular development) include innovative cycles, which are naturally pleasant for certain people.

Entrepreneurship is assumed as "the process, brought by individuals, of identifying new opportunities and converting them into marketable products or services" [33]. Northouse [34] suggests that "leadership is a phenomenon that resides in the context of the

interactions between leaders and followers and as a process; leadership can be observed in leader behaviours and can be learned". Entrepreneurial leadership is growing as a prominent leadership practice that defines a leader as a mixture of both capability and approach to meet the current market scenario's demand and gain a competitive edge over rival firms [35]. Entrepreneurial leaders work on proactive facets (such as creative, innovative behaviour) of employees. They can bring changes more significant than other leadership styles through opportunity exploration and exploitation.

Entrepreneurial leadership inculcates a conducive climate in the organization by providing autonomy that enhances autonomous motivation amongst employees and compels them to be involved in opportunities exploration and utilization. Furthermore, entrepreneurial leadership supports their team members' creative abilities to discover and exploit novel ideas [36] through autonomy, which further leads to the fulfilment of other needs, thus enhancing employees' proactive behaviour. Entrepreneurial leadership is a mixture of both entrepreneurial management and leadership orientation. These entrepreneurial leadership abilities assist organizations in sustaining their competitive edge [37]. Entrepreneurial leadership entails autonomous motivation by giving autonomy and inculcating a supportive climate that leads to ripe employees' proactive behaviour. The encouragement of proactive behaviour by entrepreneurial leadership develops the propensity to explore and utilize higher performance [38,39].

PWB is a growing research topic in the literature regarding the workplace, particularly SMEs [40]. PWB comprises self-initiating work behaviour, introducing changes, and making things happen to achieve prospects [16]. PWB can be defined as "taking the initiative in improving current circumstances or creating new ones; it involves challenging the status quo rather than passively adapting to present conditions" [41]. Proactivity is self-initiative and forward-looking action that targets revolution in self and work conditions. By concluding primarily on self-determination theory, we layout and create current conceptualizations of how proactivity is started, inspired, and advanced, adequately bringing the change. Self-determination theory gives an incredibly hypothetical focal point for clarifying how proactivity is roused. We recommend that autonomous regulation improves the probability that proactivity brings positive change for both people and associations and presents a unique model that addresses the positive upward winding of independently directed proactivity. Drawing on the arguments of self-determination theory, entrepreneurial leadership increases employees' competency by providing autonomy, thus increasing autonomous motivation to discover and exploit opportunities. Entrepreneurial leadership stimulates positive relationships among the team members and encourages an auxiliary relationship between leader and subordinate. All these efforts increase the PWB of the employees.

**Hypothesis 1 (H1).** *Entrepreneurial leadership positively influences proactive work behaviour.*

The existence of SMEs in a volatile situation [42,43] is based on the leadership and entrepreneurship competencies of their managers joined along with skills, energy, and talent [44]. Over the years, scholars have examined the skills and traits of entrepreneurial leaders [45] such as professional [46], demographic, sociological, and psychological [47] characteristics. Entrepreneurial leaders need to have appropriate skills and experience [38], especially opportunity orientation [48], creativity [49], and interpersonal skills [50], which may assist them in articulating the anticipated picture of the future, motivating followers to track their viewpoint. Self-determination theory (SDT), taken as an existing theory of psychological needs [30,51], is relevant for understanding personal thriving by reducing work uncertainty. Self-determination theory (SDT) predicates widespread instinctive emotional needs for relatedness, competency, and autonomy indicating work environments letting fulfilment of these needs assist self-initiative behaviour, psychological wellbeing, and job engagement [52] by reducing job uncertainty.

Building community and future orientation are two main characteristics of entrepreneurial leadership that distinguish them from other leadership styles. Building community refers to the endeavours of entrepreneurial leaders to inspire a favouring cast of employees

in the generation of strategic significance [53,54]. At the same time, future orientation discusses the capability of entrepreneurial leaders to articulate their foresight and head their followers in an uncertain milieu by reducing work uncertainty. Future orientation assists in decision making considering the accurate predictions related to the future. Entrepreneurial leaders affect opportunity recognition, innovation [55,56], and pro-activeness in SMEs by articulating an impressive vision, maintaining flexibility, and reducing a definite extent of uncertainty [57].

At a firm level, leading the invention process is a critical responsibility of industry leaders. A leader needs to generate a favourable atmosphere where all the employees can generate and exploit novel thoughts, contributing to innovative practices [58]. By creating a supportive climate (resulting from providing autonomy and relatedness to employees), entrepreneurial leadership encourages employees' exploratory behaviour and reduces work uncertainty. Work uncertainty may be explained as a "lack of predictability in work tasks and requirements" [59]. Uncertainty as a distinct risky element is primarily associated with makeshift employment, unemployment, or a combination [60,61]. The present workplace is fluctuating quickly [62], and the unpredictable rapid changes escalate the uncertainty level at the workplace. This work uncertainty has a destructive influence on the firm's associates and may head towards an overall deterioration of work enactment [63]. However, regardless of all theoretical and empirical evidence, there exists no empirical research investigating the intervening role of WU in the EL and PWB connection. The present study is envisioned to bridge this literary gap.

The self-determination model of work motivation guides the present study. It aims to explain how entrepreneurial leadership can relate to variables such as PWB and work uncertainty. At current work circumstances, formed by instability, insecurity, intricacy, and vagueness, workers are gradually being expected to be involved in proactive behaviour, "a set of self-starting, action-oriented behaviour aimed at modifying the situation or oneself to achieve greater personal or organizational effectiveness" [64]. Based on the argument of SDT theory, autonomy and relatedness increase the confidence level of employees. The entrepreneurial leader provides this autonomy and relatedness in autonomous motivation, which will reduce work uncertainty by improving job control and further leading to employees' proactive behaviour. Based on this, we can hypothesize that:

**Hypothesis 2 (H2).** *Work uncertainty mediates the relationship between entrepreneurial leadership and proactive work behaviour.*

The leader role in supporting proactive behaviour has been proposed and observed in multiple studies. The key evidence for this process is that experiencing help from leaders cultivates a greater sense of self-determination [65] and enhances employees' perception of willingness and competence to instigate future-focused change [66]. Numerous research has described that leader assistance positively predicts several practices of proactive behaviour, such as environmental initiative [67], personal initiative [68], creative performance [69], and idea implementation [70]. Entrepreneurial leadership is an emerging leadership style that is still in its embryonic stages of empirical and theoretical development [71]. According to leadership theory, it has emerged from the current literature of leadership and entrepreneurship [56,71]. Entrepreneurial leadership is a leadership style that encourages and motivates employees to explore and exploit business opportunities [56]. The foundational theoretical structure of entrepreneurial leadership was established by Gupta et al. [53], who presented five functions of entrepreneurial leaders, specifically, absorbing uncertainty, path clearing, framing the challenge, and specifying limits and building commitment. The first three functions are associated with situation enactment that is to visualize upcoming opportunities for entrepreneurship. The last two functions are associated with radiating enactment, i.e., to instigate and motivate team members and direct resources to achieve predetermined goals through proactive behaviour. Entrepreneurial leaders stimulate team members to become involved in proactive behaviour as an entrepreneurial leader provides autonomy

and relatedness among team members, and this autonomy and relatedness enhances the competency of the employees to take the initiative.

An essential component in proactivity is the proactive personality of employees. Employee proactivity or proactive personality can be defined as an active orientation of individuals. Proactive personality can also be defined as an impartially stable employee disposition to take individual initiative in an extensive array of activities and states [72]. However, relatedness and competency are essential for less proactive employees to obtain the needed resources to behave proactively. Instead of proactively seeking resources from diverse sources, less proactive employees incline to passively seek resources from leaders. Therefore, non-proactive or less proactive employees are likely to take advantage of leaders. Previous work on socialization (leader–member relations) outcomes is consistent with this impression [73].

**Hypothesis 3 (H3).** *The association between entrepreneurial leadership and proactive work behaviour is moderated by employees' proactive personality. Specifically, the positive relationship between entrepreneurial leadership and proactive work behaviour will be weaker when proactive personality is high.*

This research examines proactive personality as a boundary condition between the connection of work uncertainty and proactive behaviour. Proactive personality is a distinctive personality concept [74], and it refers to an individual's tendency towards proactive behaviour to bring meaningful changes in the surroundings [75]. Recent research studies exhibit the significance of proactive personality for anticipated effects such as promotions, career satisfaction, and salary [72] in addition to organizational outcomes (i.e., productivity; Kirkman and Rosen, [76]). Though plenty of research exists, research lacks proactive personality as a contingency to decrease work uncertainty and increase proactive behaviour. In a struggle to decrease the uncertainty, employees become involved in behaviours to comprehend organizational expectations and norms [77]. Proactive employees might minimize uncertainty more rapidly as three strategic qualities are related to proactivity, including being future-focused, change-oriented, and self-initiated [16]. For instance, a prominent feature of employees' proactive personality is to work on network building [78] to obtain resources relating to the job role and situation and, consequently, quickly reduce uncertainty [77]. Furthermore, a previous study has established relationships between network building, relationships, and proactive personality [78,79]. This study proposes that an inflamed linkage may facilitate the communication of firm anticipation and policies [10,77], which eventually decreases uncertainty. Based on this, we can hypothesize:

**Hypothesis 4 (H4).** *The association between work uncertainty and proactive work behaviour is moderated by employees' proactive personality. Specifically, the negative relationship between work uncertainty and proactive work behaviour will be stronger when proactive personality is high.*

This study has argued that entrepreneurial leadership behaviour of the managers is linked with employees' proactive work behaviour through work uncertainty and that proactive personality moderates the link of entrepreneurial leadership and work uncertainty. The current research adds to the literature by scrutinizing a mediated moderation mechanism of proactive personality underlying the connection between entrepreneurial leadership, work uncertainty, and proactive behaviour. Altogether, it offers a rationale for a successful mediated moderation hypothesis:

**Hypothesis 5 (H5).** *Proactive personality moderates the relationship between entrepreneurial leadership and PWB, such that the positive relationship between the entrepreneurial leadership and PWB through work uncertainty will be weaker at higher levels of proactive personality than at lower levels of proactive personality.*

The research framework of the study presents in Figure 1.

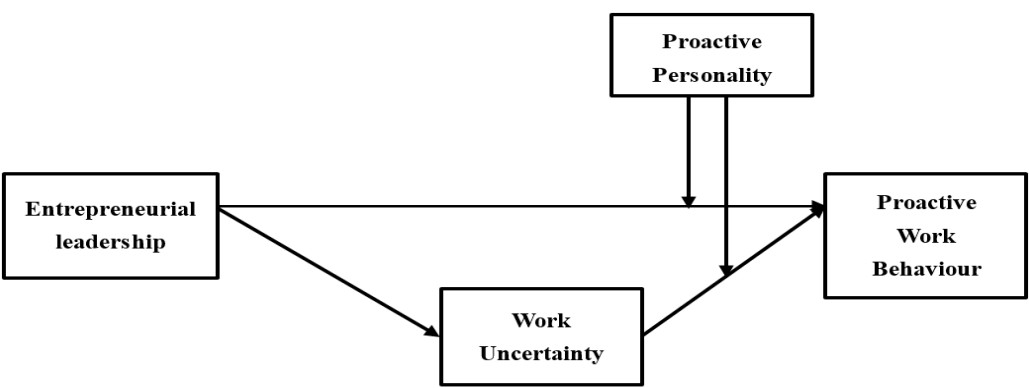

**Figure 1.** Research model.

## 3. Research Methodology

### 3.1. The Research Design

This research was materialized by utilizing the positivism research paradigm and deductive research approach. A quantitative self-administered survey questionnaire was used aligned with the research design and philosophy to gather data from the workforces of SMEs in Pakistan. The unit of analysis was individual employees and their respective supervisors. The time horizon for the current study was time lag, as data were gathered at two time points, with a gap of three weeks, to check the outcomes distinctly than predictors, and common method bias effects were reduced [80]. The time between measurements has been utilized effectively in earlier leadership studies and ought to be a lucid interval between data collections [81]. Additionally, to decrease response unfairness, all measures were not assessed by the same respondent [80]. Employees assessed all measures except PWB, which the immediate supervisor assessed in the 2nd wave.

### 3.2. Data Collection

We contacted HR managers of SMEs and took ethical permission before commencing the survey to obtain survey questionnaires filled from workers and their relevant managers. HR managers' supported us to provide a sealable envelope comprising the survey form and covering letter enlightening the tenacity and secrecy of the research. A coordinated code was utilized to recognize every worker's reaction and that of the respective manager. After the survey, all workers placed their responses inside the packet by sealing it. The replies were sent to researchers via email after collection by the HR manager.

We gathered responses utilizing two waves to decrease common method variance, with a gap of three weeks. The workers dispensed data about themselves in the first wave relating to demographics (e.g., gender, age, organizational tenure, and education). In addition, they provided data linked with their boss' entrepreneurial leadership and their own proactive personality. In Wave 2, the employees reported their work uncertainty, and the supervisors rated their employees' proactive behaviour.

In total, 800 survey questionnaires were distributed to permanent staff working in the SMEs related to high-tech industry. Initially, 651 members completed the first survey. Out of these, 502 workers completed the succeeding survey after three weeks. In Wave 2, we also distributed questionnaires to the corresponding supervisors of the employees who responded in Wave 1 and obtained 453 supervisor responses. Because of missing information, unequalled codes Time 1 (T1) and Time 2 (T2), and low effortful responses (inadequate surveys were detached), 33 members' files were removed, resulting in a concluding sample of 420 respondents answering the three surveys.

Out of the complete sample, 68% were male, and 32% were female; regarding education, 34.4% had less than 14 years of education, 18.5% possessed 14 years of education, 37.1% had 16 years of education, and 10% had 18 years or more than 18 years of education. The sample consisted of relatively young employees: 34.3% were 25 years or less than

25 years, 21.4% were between 26 and 30 years, 30% were between 31 and 35 years, 15.9% were between 35 and 45 years, and 1.5% were above 45 years. All measures were assessed on a five-point Likert scale from 1 strongly disagree to 5 strongly agree. Work uncertainty and proactive work behaviour were also measured on a five-point Likert scale with little variation from scale 1 = not at all to 5 = a great deal and 1 = very rarely to 5 = very frequently, respectively. Measures used to assess each variable are presented in Table 1.

**Table 1.** Study variables measures.

| Variables | Sample Items | References |
|---|---|---|
| Entrepreneurial leadership: 8-items scale | "My team leader has creative solutions to problems" and "My team leader demonstrates a passion for his/her work". | Renko et al. [56] |
| Work uncertainty: 9-items scale | "Does the equipment you use work reliably"? And "Do you come across unexpected problems in your work"? | Leach et a. [82]. |
| Proactive personality: 10-item scale | "I excel at identifying opportunity" and "I am always looking for better ways to do things". | Seibert et al. [72] |
| Proactive work behaviour: 13-item scale | "How frequently do you promote and champion ideas to others"? And "How frequently do you try to institute new work methods that are more effective"? | Parker and Collins [11] |

### 3.3. Data Analysis

A statistical software package SPSS 20 was used to analyse the data. Additionally, to estimate the relationship among study variables, correlation analysis was used, displayed in Table 2. Hierarchical regression analysis was applied for hypotheses testing and to check the direct effects of an independent variable (i.e., entrepreneurial leadership) on an outcome variable (i.e., proactive work behaviour). Baron and Kenny's [83] prescribed method was used for checking the mediating effects of work uncertainty for the connection between entrepreneurial leadership and proactive work behaviour. Preacher, Rucker, and Hayes' [84] moderated mediation test steps were applied at the end of the study.

**Table 2.** Descriptive statistics, reliability, and correlation scores.

| Variables | Mean | S.D. | 1 | 2 | 3 | 4 | 5 | 6 | 7 | 8 |
|---|---|---|---|---|---|---|---|---|---|---|
| 1. Gender | 0.69 | 0.46 | | | | | | | | |
| 2. Age | 2.30 | 1.11 | 0.10 * | | | | | | | |
| 3. Qualification | 2.23 | 1.03 | −0.33 | −0.09 | | | | | | |
| 4. Experience | 1.61 | 0.87 | 0.09 | 0.66 ** | −0.27 ** | | | | | |
| 5. M_PWB | 3.73 | 0.49 | −0.15 | — | −0.14 ** | 0.06 | (.73) | | | |
| 6. M_EL | 3.80 | 0.67 | −0.06 | 0.26 ** | 0.15 ** | 0.07 | 0.35 ** | (0.85) | | |
| 7. M_PP | 3.91 | 0.58 | −0.10 * | 0.14 ** | 0.27 ** | −0.14 ** | 0.37 ** | 0.63 ** | (0.88) | |
| 8. M_WU | 2.16 | 0.57 | −0.048 | −0.29 ** | −0.05 | −0.14 ** | −0.54 ** | −0.80 ** | −0.58 ** | (0.79) |

Notes. Age, qualification, and tenure of the employees is in years; Cronbach's alpha reliability scores are reported in parentheses, * $p < 0.05$, ** $p < 0.01$.

## 4. Results

The descriptive statistics, reliability scores, and correlation between all the key variables of the study are presented in Table 2. The correlation between all the key variables of the study was statistically significant; thus, it provides initial evidence for all hypothesized relationships of the study. In the control variables, except age (which showed no association with proactive work behaviour), all the other demographics have relationships with the study variables, shown in Table 2. Next, in the hypotheses testing assumption, the study's first hypothesis posited that entrepreneurial leadership possesses a positive association with the proactive work behaviour of the employees.

The results shown in Model 1 of Table 3 demonstrated that entrepreneurial leadership was positively related to proactive work behaviour (β = 0.21, $p$ < 0.00), thus providing empirical backing for H1 of the study. Further, the study's second hypothesis claimed that work uncertainty mediates entrepreneurial leadership and proactive work behaviour. For testing mediation effects, we have used the steps of Baron and Kenny [83] to check mediation effects of work uncertainty for the association between entrepreneurial leadership and proactive work behaviour. The independent variable must be associated with the dependent variable (H1 of the study) to confirm the mediation effect. Second, the relationship between the independent and mediating variable should be statistically attested, as shown in Model 2 of Table 3. Third, the mediator must be related to the outcome variable. In the last step, upon entering both the independent and mediator together in the regression equation, either the independent variable becomes insignificant (full mediation), or it drops in its effect but is still significant (shows partial mediation). Results mentioned in Table 3 explained that entrepreneurial leadership (independent variable) is significantly related to proactive work behaviour (outcome variable), H1 of the study. Further, shown in Model 4 of Table 3, entrepreneurial leadership is also significantly related to work uncertainty (β = −0.67, $p$ < 0.000). Next, work uncertainty, the mediator, is also associated significantly with proactive work behaviour (β = −0.53, $p$ < 0.01).

**Table 3.** Hierarchical regression results.

| Variables | PWB | | | WU | PWB | |
|---|---|---|---|---|---|---|
| | Model 1 | Model 2 | Model 3 | Model 4 | Model 5 | Model 6 |
| Gender | −0.22 *** | −0.26 *** | −0.28 *** | −0.10 ** | −0.26 *** | −0.12 *** |
| Age | −0.07** | −0.09 *** | −0.09 *** | −0.02 | −0.1 *** | −0.67 *** |
| Education | −0.12 *** | −0.12 *** | −0.11 *** | 0.01 | −0.13 *** | −0.12 *** |
| Tenure | 0.05 | −0.04 | −0.03 | −0.03 | 0.06 | −0.03 |
| EL | 0.3 *** | | −0.15 ** | −0.67 *** | 0.07 | −0.23 *** |
| WU | | −0.53 *** | −0.67 *** | | 0.61 *** | −1.5 *** |
| PP | | | | | −0.34 * | 0.31 * |
| WU*PP | | | | | | 0.22 *** |
| EL*PP | | | | | 0.08 | |
| Adj R$^2$ | 0.21 | 0.4 | 0.41 | 0.65 | 0.44 | 0.46 |
| F Value | 22.79 *** | 57.05 *** | 50.27 *** | 17.54 *** | 42.56 *** | 46.39 *** |
| ΔR$^2$ | 0.15 | 0.34 | 0.36 | 0.56 | 0.45 | 0.41 |

*** $p$ < 0.001, ** $p$ < 0.01, * $p$ < 0.05.

When entrepreneurial leadership and work uncertainty were entered together in the regression model, the value of entrepreneurial leadership reduced from 0.30 to −0.15 (comparing Model 1 and Model 3 of Table 3) but significantly highlighted that work uncertainty partially mediated the association of entrepreneurial leadership and proactive work behaviour. Further, the Sobel test was carried out to assess the mediation effect of work uncertainty and found it significant ($p$ < 0.001). The achieved results empirically supported the H2 of the study. The third hypothesis of the research stated that proactive personality moderates the relationship between entrepreneurial leadership and proactive work behaviour. To gauge moderation and mediated moderation outcome [85], entrepreneurial leadership must be significantly associated with proactive work behaviour, which is already assessed as presented in Model 1 of Table 3. Secondly, the scholars tested whether the interaction among entrepreneurial leadership and the proposed moderator (i.e., proactive personality) is significantly associated with proactive work behaviour [85]. Outcomes linked to the interaction of entrepreneurial leadership and proactive personality on proactive work behaviour is presented in Table 4. The results exposed that proactive personality as a moderator interacted with entrepreneurial leadership to forecast employees' proactive work behaviour. The less proactive personality employees are more inclined to show proactive work behaviour, as shown in Figure 2. The result is consistent with the previous research [73,86]. Therefore, Hypothesis 3 of the study was also supported by the data.

**Table 4.** Index of mediated moderation.

| PP | PWB | | | |
|---|---|---|---|---|
| | B | SE | Z | P |
| −1 SD | 0.56 | 0.04 | 3.33 | 0.000 |
| M | 0.42 | 0.03 | 3.91 | 0.000 |
| +1 SD | 0.28 | 0.06 | 4.49 | 0.000 |

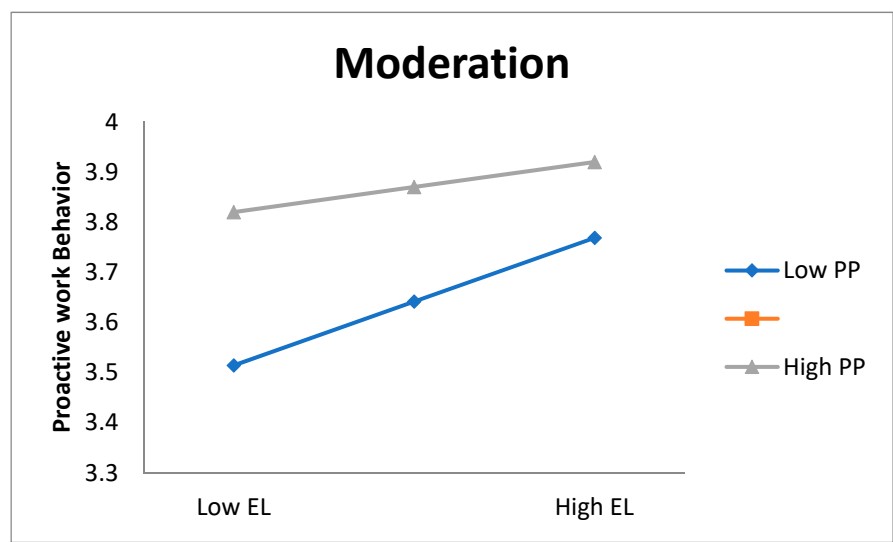

**Figure 2.** Moderation relationship plot. EL = entrepreneurial leadership, PP = proactive personality.

The fourth hypothesis described that proactive personality moderates the relationship between work uncertainty and proactive work behaviour. According to Preacher et al. [84], work uncertainty must be significantly related with proactive work behaviour, which is already evaluated, as presented in Model 2 of Table 3. Secondly, the interaction among work uncertainty and proactive personality is significantly related with proactive work behaviour [84], as shown in Figure 3. Results related to the interaction of work uncertainty and proactive personality on proactive work behaviour are presented in Table 3. The results revealed that proactive personality as a moderator interacted with work uncertainty to forecast employees' proactive work behaviour. Hence, Hypothesis 4 of the study was also supported by the data. For Hypothesis 5, the condition for mediated moderation, suggested by Preacher et al. [84], is already supported, i.e., entrepreneurial leadership is significantly related to proactive work behaviour (Model 1 of Table 3). To obtain further confirmation, researchers examined that the extent of the conditional indirect outcome of entrepreneurial leadership via work uncertainty was different at low, average, and high levels of employees' proactive personality on proactive work behaviour. For this, we used Preacher et al.'s [84] statistical significance test, where a z statistic was checked to judge the conditional indirect effect. Further, we operationalized low, average, and high levels of the moderating variable (i.e., proactive personality) as one standard deviation (SD) below and above the mean score. The estimates, z statistics, common errors, and significance values of conditional indirect effects are shown in Table 4. The results exposed that the indirect effects of entrepreneurial leadership and uncertainty were significant at a lower level for the moderator variable (proactive personality). The plot of the interaction effect was also documented in Figure 2. Consequently, Hypothesis 5 of the research is also empirically proved.

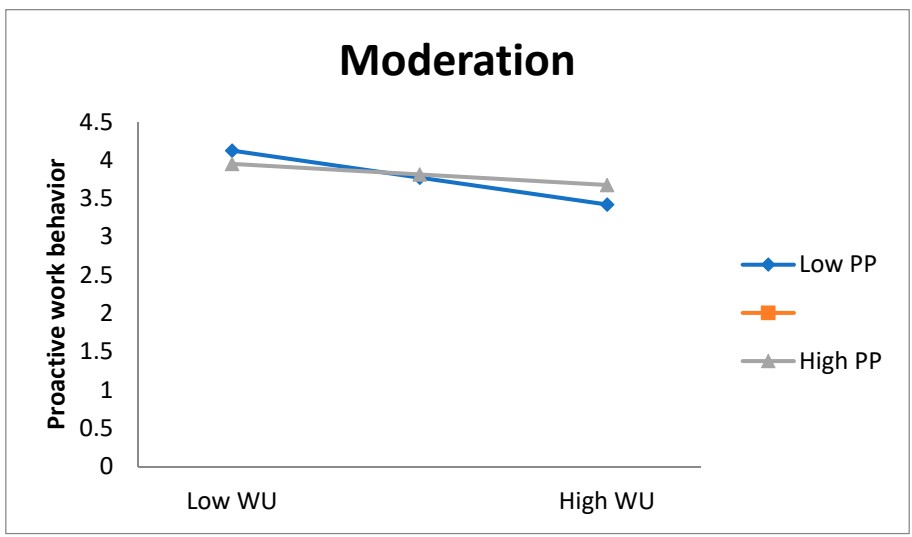

**Figure 3.** Moderation relationship plot. EL = entrepreneurial leadership, PP = proactive personality.

## 5. Discussion

Theoretically, the research supplements the academic literature in several ways. The study widens our understanding of the positive effect of leadership on employees' behaviour [87], such as proactive work behaviour. Moreover, the study supplements the leadership–proactive behaviour literature by discovering the mediating effect of work uncertainty between entrepreneurial leadership and PWB, and previously, no research exists. The research also showed the significance of proactive personality as a moderator among work uncertainty and PWB and between entrepreneurial leadership and PWB. It is an exceptional study highlighting the importance of proactive personality (as a boundary condition) in weakening the relationship between entrepreneurial leadership and PWB, making a unique contribution to the self-determination theory and leadership–proactive behaviour literature. Furthermore, high proactive personality minimized the effect of work uncertainty and increased the proactive behaviours of employees. The research also presents entrepreneurial leadership as a needed style for nurturing PWB compared to the well-established transformational leadership and successfully displays diverse results concerning encouraging employees' PWB. Practically, the outcomes of the research can be exploited by the leaders for stimulating workers' PWB. The researchers suggest adopting a positive leadership style for encouraging the positive behaviour of employees [88] such as PWB. The results display that an entrepreneurial leader encourages followers to focus on exploration and exploitation [89] by instilling self-assurance in employees to take risks and attempt new ideas leading to proactive behaviour at the workplace [90]. Entrepreneurial leaders allow adequate space to the followers for exploration at work [91], which resultantly encourages employees' PWB. The sentiment of reciprocation instilled by an entrepreneurial leader [89,91] also encourages followers to behave proactively [92] and develop inventive ideas and suggestions for backing toward the growth of the business.

Moreover, the study reiterates how entrepreneurial leaders minimize work uncertainty in establishments [91], which encourages PWB. In the first place, an entrepreneurial leader reduced work uncertainty by presenting himself as a model by freely exploring and exploiting opportunities in the organization [93]. Entrepreneurial leaders encourage the participation of workers in decision making and prompt followers to collaboratively solve problems [89,94], creating a nurturing climate not bothered by considering work uncertainty as a hurdle and, instead, viewing it as an opportunity to excel in the organization. Distinctively, the bearing of proactive personality in boosting PWB in the business is recognized by the researchers. The proactive personality fosters employees' control over their localities. It boosts their self-confidence [95]. This self-confidence reduces dependency on the leader to obtain needed resources to behave proactively. Therefore, less

proactive personalities show more proactive behaviour in the presence of entrepreneurial leadership [79], as consistent with the previous research [73]. However, in work uncertainty, proactive personality encounters the uncertain work condition and behaves more proactively than less proactive personality.

## 6. Conclusions

The research enriches our understanding of the influence of entrepreneurial leadership on PWB. The mediated mechanism through the moderating effect of PP and mediating impact of WU offers rich insights on how leaders today can enhance the proactive behaviour of followers by assisting them in exploration and exploitation and decreasing work uncertainty amongst them. The research further comprehends the significance of PP, which moderates the association between entrepreneurial leadership and PWB and affects the indirect relationship between entrepreneurial leadership on PWB. The leaders may use the research findings to decrease work uncertainty at the workplace by encouraging entrepreneurship among employees for the proactive work environment.

*Limitations and Future Recommendations*

The research is subject to numerous limitations. The cross-sectional method of the research bounds the researchers from establishing the causality among entrepreneurial leadership and PWB. Longitudinal research may be conducted to explain more conclusive overviews and study whether an entrepreneurial leader enhances PWB, since long seeing the anticipated behavioural fluctuations of the lead. Future research is encouraged to inspect the association between entrepreneurial leadership and PWB by seeing further probable mediators, for instance, employee ambidexterity, which has been suggested as the outcome of entrepreneurial leadership. Similarly, other latent moderators may be measured rather than proactive personality to inspect the link between entrepreneurial leadership and PWB. The forthcoming study may also explain the comparison between entrepreneurial leadership and transformational leadership and their impact on PWB in the organization.

**Author Contributions:** Conceptualization, M.B. and S.C.; methodology, S.A.; formal analysis and investigation, M.S.; writing—original draft preparation, S.A. and H.A.; project administration and supervision, S.C.; writing—original draft preparation, K.S. All authors have read and agreed to the published version of the manuscript.

**Funding:** This research received no external funding.

**Institutional Review Board Statement:** Not applicable.

**Informed Consent Statement:** Not applicable.

**Data Availability Statement:** Data are available and can be provided on request.

**Conflicts of Interest:** The authors declare no conflict of interest.

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
