# Peer review of "Entrepreneurial Leadership and Employees’ Proactive Behaviour: Fortifying Self Determination Theory"

_2199-8531, doi:10.3390/joitmc7030176_

Round 1
Reviewer 1 Report
The study deals with entrepreneurial leadership, proactive behaviour and job uncertainty. Overall, the authors demonstrate a good scientific soundness in the article. The article has a good introduction and makes clear the objectives, develops the hypotheses well, the methodology is suitable and reflects well the results and conclusions, however, in the exposition and structure of the methodology there are aspects that could be improved.
Firstly, in the summary, I would eliminate more introductory aspects and make clear the type of methodology, which is quantitative research by means of a questionnaire.
Regarding the structure of the methodology, I would follow a structure with the following sections: research design, data collection and data analysis. I think that with this structure of methodology it would be much clearer for the authors what type of research they have proposed and developed, since, in my opinion, the information is not well ordered, which could be solved by following this structure, making it clear what type of research has been developed, how the survey has been carried out, how it has been sent to the sample, how it has been analysed, etc.
To facilitate the information in the questionnaire, it would be ideal to make a table specifying the measurement variables, the measurement items and the reference taken into account.
As I have stated before, the good layout of the methodology is, in my opinion, the point on which the authors should work to improve the work, since the rest of it is considered to be well implemented and of an adequate scientific quality.
Author Response
We are thankful to you for the constructive and positive comments on our paper. We have thoroughly revised the paper according to your suggestions, and we believe that they have helped us significantly improve the quality of the current version of the document. We outline below the amendments that we have made in response to each of your comments. Moreover, the detailed response sheet on reviewer A comments is attached herewith.

Reviewer 2 Report
The article is well-written and so is the presentation of the results. I would like to see a more general introduction (historical and sociological) so as to attract more interest from the readers.
Also some more details about the country's experience and specificities with the Pandemic. Some more discussion about the experience from other countries with regard to similar studies. So the reader can judge better the generality of the results.
Author Response
We want to thank you for the constructive and positive comments, which have helped us significantly improve the current version of the paper. We outline below the amendments that we have made in response to each of your comments. Moreover, the response sheet on Reviewer B comments is also attached herewith.
